# Causality can systematically address the monsters under the bench(marks)

## Abstract

Effective and reliable evaluation is essential for advancing empirical machine learning. However, the increasing accessibility of generalist models and the progress towards ever more complex, high-level tasks make systematic evaluation more challenging. Benchmarks are plagued by various biases, artifacts, or leakage, while models may behave unreliably due to poorly explored failure modes. Haphazard treatments and inconsistent formulations of such "monsters" can contribute to a duplication of efforts, a lack of trust in results, and unsupported inferences. In this position paper, we argue causality offers an ideal framework to systematically address these challenges. By making causal assumptions in an approach explicit, we can faithfully model phenomena, formulate testable hypotheses with explanatory power, and leverage principled tools for analysis. To make causal model design more accessible, we identify several useful Common Abstract Topologies (CATs) in causal graphs which help gain insight into the reasoning abilities in large language models. Through a series of case studies, we demonstrate how the precise yet pragmatic language of causality clarifies the strengths and limitations of a method and inspires new approaches for systematic progress.

## 1 Introduction

Machine learning achievements continue to break records and grab headlines, drawing attention from both the public and the research community. However, the rapid proliferation of powerful models and the increasing complexity of tasks continue to amplify existing challenges in reliable evaluation of these models [1]. Between inflated expectations [2–4], opaque or misleading assessments [5], and even the occasional mistake [6], the poor communication [7] and unreliable benchmarks [8–10] can significantly undermine our understanding of the capabilities and limitations of these models [11, 12]. This risks a decline of public trust [13–15] and perhaps even an AI winter [16]. A key issue is that many evaluations focus on performance alone [17], failing to account for the reasoning process behind a model's behavior. For instance, a model may arrive at the right answer for the wrong reasons, making the performance alone an incomplete indicator of its capabilities.

To systematically address the challenges in evaluating, in particular, large models, **this position paper argues for a shift toward causality-driven experimental design.** By making causal assumptions explicit, we formulate precise hypotheses and underlying assumptions, diagnose model limitations, and leverage principled tools for analysis.

One subfield that is particularly well-fitted for more causal analyses is the evaluation of reasoning abilities in large language models (LLMs) [18, 19]. A cursory analysis of the recent NLP papers in the ACL anthology reveals a dramatic rise in the attention to the reasoning capabilities of models, as seen in Figure 1a. However, curiously, the subset of these papers that mention "causality" or "causal" in the title or abstract is not growing in tandem (yet). In fact, the dendrogram in Figure 1b shows

Submitted to 39th Conference on Neural Information Processing Systems (NeurIPS 2025). Do not distribute.

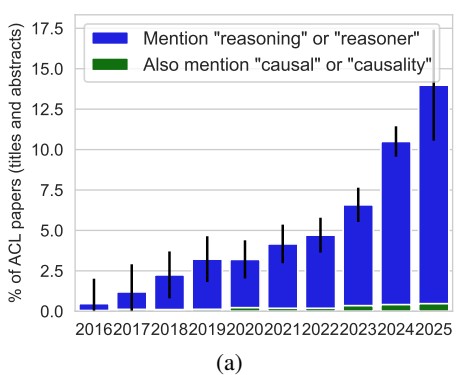 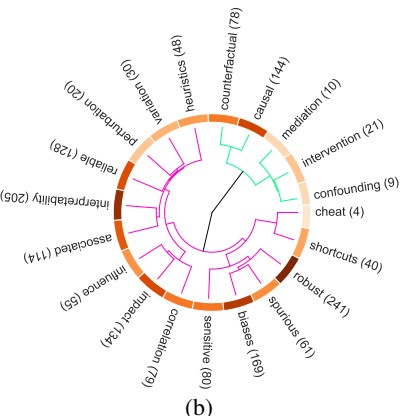

(a)            (b)

Figure 1: (a) Growth of reasoning papers in the ACL Anthology, among which the concept of "causality" is not growing at the same rate, suggesting that NLP is underutilizing causality. (b) This dendrogram shows the co-occurrences of causal and causality-adjacent terms of papers that contain "reasoning" in the abstracts (total 3181 papers) from the ACL anthology from the past 10 years. The numbers in parentheses indicate the number of papers that mention the term. Note, that the very first split separates all the causality-related terms from the rest of the terms, suggesting relatively poor co-occurrence with other related concepts.

that among the reasoning papers, causality-related terms tend not to co-occur very much with many non-causal mimics (discussed in Section 2).

Although these issues often appear disparate, we argue that causality can serve as the framework to systematically study a wide array of issues thus bridging gaps between different subfields and approaches. The expertise required to understand and satisfactorily mitigate these issues is very diverse, and we do not mean to suggest that causality is *all* you need. Rather, we argue that causality already (often implicitly) underlies much of the design, analysis, and interpretation of machine learning experiments, despite the apparent gap in terminology as suggested by Figure 1b.

To make an explicit causal framing more accessible and attractive, we develop a guide to common causal structures and then use these abstract topologies to gain insights across case studies about reasoning abilities in LLMs. While we focus on research questions and issues concerning the evaluation of reasoning abilities in LLMs, all four of our main claims (particularly 2-4) largely apply to the whole of empirical machine learning research. Our contributions include:

- We survey a wide variety of common challenges, or *monsters*, that hinder reliable evaluation and discuss how these can faithfully be modeled using causality
- We propose intuitive templates for causal modeling, called Common Abstract Topologies (CATs), and show how these aid in effective model design
- We motivate practical causal models to approaches in several case studies
- We promote causality as a tool to encourage hypothesis-driven research with precise, transparent assumptions and more systematic evaluation

## 2 A Bestiary of Monsters

As LLMs become an increasingly integral component of decision-making systems, reasoning failures (including seemingly toy problems like in Example 1) may have dire real-world consequences [11, 13]. This necessitates more powerful tools to identify, catalog, and address the bestiary of issues that arise in the design as well as evaluation of large models.

Example 1 is indicative of several common reasoning failures in LLMs, and, by implication, our evaluation procedures for addressing such issues. The input prompt is structurally similar to the questions found in GSM8k [20], a benchmark used for evaluating a model's multi-step arithmetic and commonsense understanding skills. However, we select the subject matter and word choice to evoke a mobster discussing plans to extort money from local businesses. Although the LLM's final answer is factually correct, there are several problems with the rationale: (1) it makes several arithmetic

> **Prompt:** Listen here, I'm starvin' and it's already half past four. Gotta rake in 10 grand in protection dough before I eat. I'm hittin' up every joint in the hood with my bat, collecting an average of 800 bucks a pop. Takes me 10 minutes to move from one mark to the next. Am I gonna get to chow down before 7?
>
> **LLM Chatbot:** If you're aiming to collect 10 grand by **shaking down joints** for an average of $800 each and it takes you 10 minutes to move from one target to the next, it would take you approximately **3 hours and 20 minutes** (200 minutes) to reach your goal. Starting at half-past four, you should be able to hit your target before 7 PM. So, **yes, you can have a meal before 7** if everything goes as planned. **Enjoy your dinner!**

Example 1: **The final answer is correct, and yet the reasoning is very wrong.** Although this LLM's response is technically correct in a narrow sense, its answer for this GSM8k-like multi-step reasoning question [20] is unsatisfactory for the purposes of evaluating the model's arithmetic reasoning, as well as commonsense or moral reasoning as the LLM appears not to pick up on the obvious unethical context implied by the scenario.

mistakes which happen to cancel out, (2) it fails to pick up on the unethical situation implied by the scenario, and (3) by implicitly condoning the criminal behavior, it does not consider the broader consequences of the response. Crucially, if we only check for correctness, as is standard practice [18], we would find no fault in the response.

The problem is that to demonstrate good reasoning abilities, a correct answer is insufficient. We need to show that the model answers the question correctly *for the right reasons*. In other words, our evaluation must verify that the model's processing of the input information *leads to* the correct answer consistently and reliably. This criterion makes a *causal* claim about the model's reasoning process, and thus must be supported by a causal analysis.

> **Claim 1:** Evaluating reasoning involves causal inference
>
> A correct answer can be reached through very poor reasoning, but poor reasoning will not generalize beyond the lab bench. To generalize well, the model's reasoning must rely on robustly predictive (i.e. causal) features and relationships rather than spurious ones. Consequently, to meaningfully evaluate reasoning abilities, one must assess what influences *how* the model arrives at its predictions, which is inherently a causal inference problem.

## 2.1 "Here be dragons" [1]

To get a qualitative sense of the myriad of issues, or *monsters*, that plague our benchmarks and experiments, we will briefly survey recent approaches, including broad overviews into the nature of reasoning tasks [18, 19] and the evaluation of LLMs [1, 21, 22]. For investigations of more specific issues, we separate efforts into three clusters depending on whether the problem originates with the (1) models, (2) datasets, or (3) evaluation procedures.

**Models**    This line of work focuses on characterizing the reasoning failures and biases of language models, which is nontrivial given their opaque behavior [23]. These failures range from well-defined formal errors such as logical fallacies [24], red herrings [25], or invalid inferences [26] to broader issues including sensitivity to superficial features [3, 22], overconfidence [11], hallucinations [27, 28], and lack of robustness [29–31]. Some studies explore how models exhibit "content effects" [32], absorbing and amplifying human biases [33, 34] including social and cultural biases [13, 35–40], such as stereotyping [41].

**Datasets**    Meanwhile, subtle variations of popular benchmarks, such as premise order in reasoning tasks[42] or minor changes in problem parameters [43, 44], can cause large performance drops [11, 12], raising concerns not just about whether models genuinely reason [45], but also about exploitable issues in the training data and benchmarks [9, 46]. These can be described as enabling cheating [47],

---

[1]The vague and uneasy language researchers often use when alluding to biases or unresolved limitations in their evaluations is reminiscent of how medieval cartographers would fill the unknown edges of their maps with dragons.

heuristics [48, 49], or shortcuts [50–52], possibly due to sampling biases [53] or in certain cases even leakage between the training and testsets [47] which can result in memorization [54]. Poor dataset construction can lead to annotation artifacts [55, 56] such as priming effects [57], which degrade the quality and reliability of results [58] while also unintentionally reinforcing social biases [59] or cultural inequities [13, 60, 61].

**Evaluation**    Even with well-constructed datasets, evaluation methodologies can introduce systematic errors [62] or lead to misleading conclusions [7]. For example, automated scoring systems can obscure obvious failures [6], while static benchmarks can emphasize surface-level accuracy at the cost of other important factors, such as generalization [17] or interpretability [63] or social costs [8, 13]. While standardized leaderboards [64] and evaluation procedures [65] can enable more direct model comparisons, these benchmarks can gradually become less representative of real-world tasks [10, 66–68], introduce biases that favor certain model families [69], or inadvertently leak information from the test set [47] which can be difficult to detect due to closed-source models and proprietary datasets [1].

Despite the diverse, at times redundant, terminology, we observe certain structural similarities in the approaches of these contributions. Terms like "ablation", "perturbations", "edits", "flips", "masking" can often be interpreted as interventional or counterfactual analyses, while "sensitivity"/"robustness", "consistency", "shortcut", "leakage", "bias", etc. refer to how the model's behavior is impacted by, for example, (seen or unseen) confounders.

> **Claim 2:** The monsters are causal
>
> Many of the recurring issues in benchmark evaluation, including biases, spurious correlations, or systematic failure modes, are often described in vague or ad hoc terms. However, these issues arise from specific causal features of the underlying data-generating process or evaluation procedure. Whether the factors are known or latent, their influences can be captured by an appropriate causal model to formulate precise, testable hypotheses and guide more principled experimental design.

# 3   Common Abstract Topologies

| Name | Graph | Example Phenomena |
|---|---|---|
| Confounding |  | • prompt wording, instruction tuning, or prompting strategies
• dataset sourcing, annotation artifacts, missing context
• overlap or leakage between the benchmark and training data |
| Mediation |  | • circuit analysis such as mechanistic interpretability
• tool use or integrating an LLM in a larger application
• editing individual tokens or ablating model parameters |
| Spurious Link |  | • social and cultural biases in the data collection process
• imbalances in the surface form such as symbol or label bias
• variable selection and construction |

Table 1: Some simple Common Abstract Topologies (CATs) used to formalize a wide variety of *monsters* that may lurk in benchmark or experiment analysis. The graphs use ▨ for the independent variable, ▨ for the dependent variable, and ⬣ for a third-variable factor.

Creating a causal graph that faithfully represents the underlying structure of an experiment or data generating process can be very challenging. Especially since, when we design an experiment, we usually think in terms of more vague concepts like independent, dependent, and controlled variables, and consequently only implicitly make causal assumptions. However, explicit causal graphs:

- precisely communicate the assumptions that go into a benchmark, experiment, or analysis
- leverage the machinery of causal inference for more principled analyses
- understand the implications of our design choices including the strengths and limitations on both technical and conceptual levels

To help make the process of constructing a causal graph more accessible and systematic, we identify a set of Common Abstract Topologies (CATs) that frequently appear in causal graphs. For motivation, we list some associated phenomena (see Table 1) in the context of evaluating reasoning abilities in large models, where these patterns may offer useful abstractions.

However, researchers may be hesitant to commit to a specific causal graph that fully captures all factors influencing their analysis [70]. In practice, causal graphs are often underdetermined by the available data and may hinge on subtle choices in how variables are defined or interpreted [60]. As Loftus [63] point out, some researchers even avoid causal framing altogether, since it makes explicit assumptions that reviewers may challenge.

> **Claim 3:** Instrumentalism is all you need for model design
>
> A causal model does not need to be perfect to be useful. Plausible simplifying assumptions and abstractions can yield valuable insights and motivate practical experiments. As research advances, the model can be incrementally refined, while providing precise falsifiable hypotheses at every step of the way.

We join Loftus [63], Janzing and Garrido [71] in advocating for a more pragmatic, instrumentalist attitude to causal modeling. In many cases, the same phenomenon can be represented by multiple causal graphs that differ in variable selection, construction, or level of abstraction [72–74]. Nevertheless, as long as a proposed causal model does not directly conflict with the available data, it may be sufficient to produce actionable insights (such as more interpretable or explainable models).

Aside from the additional explanatory power, if a more formal treatment is necessary or desired, there is a vast world of tools and techniques to explore. The field of causal inference [70, 75–78] has developed a language for formalizing the effects of subtle design choices and their, potentially counterintuitive, consequences for the analysis. For example, Simpson's paradox can be elegantly explained, to "resolve" the apparent paradox based on the appropriate causal assumptions of the problem (for a deep dive into this topic see Pearl [79] and Chapter 6 of Pearl [76]).

> **Claim 4:** Towards *explicit* causal assumptions
>
> An experimental design involves a variety of assumptions about what factors matter, how they interact, and how this relates to the proposed approach. Here the language of causality provides a powerful framework for motivating an approach, precisely formulating the hypothesis, and answering questions in a principled way.

Causal inference is valuable not only for formal analysis but also as a conceptual framework for understanding the structural assumption behind an approach or argument. By making the concepts and tools of causal inference more accessible, we aim to develop a practical guide to recognize familiar causal structures in the real world, as well as build an intuition for the implications of model design choices on analysis and interpretation. To this end, we present three simple CATs that correspond to the three causal interpretations of a statistical dependence between two variables according to Reichenbach's common cause principle [80].

Here is a brief sketch of how CATs can be used to guide model or benchmark design:

1. **Identify a relationship of interest**: Select a measurable outcome variable (e.g., model accuracy) and a primary explanatory "stimulus" (e.g., input prompt, fine-tuning data, or model parameter).

2. **Enumerate additional influences**: Consider other factors likely to affect the outcome either directly or indirectly. Assess whether they are conceptually *upstream* (potential confounders), *downstream* (potential mediators), or *parallel* sources of variation (potential sources of spurious correlations) with respect to the stimulus. Based on these relationships, consider the respective CAT(confounding, mediation, or spurious links) to serve as a structural template.

3. **Refine the graph**: Adapt the structure based on available data and the specific research question. Variables and edges may be omitted or aggregated, provided the resulting model supports plausible hypotheses and does not contradict observed dependencies.

4. **Use the model to guide analysis**: Apply the graph to derive testable implications (e.g. identify estimable causal queries by causal inference), suggest experimental interventions, or motivate mitigation strategies (e.g., balancing, ablation, or regularization).

### 3.1 Confounding

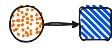

Confounding occurs when there is a common cause between the stimulus and response variables. For our purposes, we further restrict the "confounding" CAT to the case where the confounder is known and can, in principle, be controlled for. Confounding makes evaluation difficult or unreliable because the observed statistical relationship between the stimulus and response is not representative of the underlying causal relationship, thus unbiased causal effect estimation necessitates controlling for the confounder.

### 3.2 Mediation

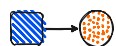

Another important type of causal topology is mediation, where there are multiple causal paths between the stimulus and response. For simplicity, we illustrate this general structure with one direct causal link and one that goes through a mediator variable. Mediation analysis is often used to quantify the impact of subcomponents or side-effects on the model's behavior. For example, a common setting may be to study the impact of a specific prompting strategy or representation on the model's response, which can be modeled as mediation as in Figure 2.

The impacts of the individual causal paths can be studied by estimating the natural direct effect (NDE), natural indirect effect (NIE), or controlled direct effect (CDE) [75]. However, notably controlling for the mediator is not always appropriate, for example, for estimating the total causal effect (TCE). This underscores one of the key benefits of causal inference: given the specific causal query, the appropriate analysis method is dictated by the graph structure, thereby prescribing specific and principled experiments.

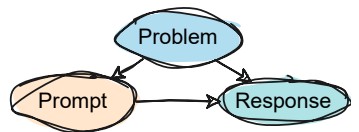

Figure 2: Sketch of a conceptual causal model treating the prompt (i.e. surface form) as a mediator between the underlying problem or task of interest and the model's response.

### 3.3 Spurious Links

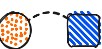

The third Common Abstract Topology (CAT) we discuss here describes a variety of graphical structures that give rise to spurious (i.e. non-causal) associations. More specifically, spurious links are statistical dependencies between variables that are not causally related (neither is an ancestor of the other), but are correlated due to an unblocked non-causal path in the underlying graph. This is often due to (1) a common cause (a confounder), which is usually unknown or too complex to be modeled explicitly, or due to (2) a common effect (a collider) which is conditioned on, thereby activating a backdoor path [75]. We denote non-causal links in a causal graph with a dashed curved edge, indicating that the observed association lacks a direct causal explanation.

Crucially, although spurious links are non-causal, models can still learn to exploit them for prediction, thereby effectively introducing an undesirable dependency (denoted by a dashed grey arrow) between the spurious feature and the outcome. This CAT is particularly relevant in machine learning, where models are typically trained on observational data only. In the absence of interventional or counterfactual signals, there is no principled way to distinguish causal from non-causal associations, making it easy for models to rely on spurious correlations in the data without appropriate inductive biases.

Another common source of spurious links, particularly in datasets, is due to selection bias, where the dataset includes only a subset of instances based on some unmodeled criteria. This selection process acts as a collider and can introduce spurious associations between otherwise independent variables.

Generally, it is not feasible to entirely eliminate spurious links, as seemingly innocent choices in variable construction and selection are invariably informed by the experimenter's biases [60, 81]. Nevertheless, there is extensive causal inference machinery to address spurious correlations depending on the specific setting [82].

## 4 Case Studies

In this section, we discuss a variety of specific research projects which either make use of one of the Common Abstract Topologies (CATs) or could benefit from a more *explicitly* causal framing.

### 4.1 Confounding

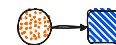

Several recent projects have used causal framing to identify or address confounding in LLM behavior. For example, Xia et al. [83] propose using a reward model as an instrumental variable to control for confounding in prompt biases. In a similar setting, Hüyük et al. [84] generate counterfactual samples to improve causal consistency. In human-LM collaboration, Zhang et al. [85] treat prior human and model actions as confounders influencing future decisions and introduce the Incremental Stylistic Effect to measure interventions in multi-turn interactions.

Meanwhile, an active area of research which could effectively be understood using the confounding CAT, investigates how dataset artifacts affect the performance of LLMs on mathematical reasoning tasks, such as GSM8K [20]. These studies vary premise order, subject matter, or input distributions [42, 43, 45, 69], often uncovering unexpected sensitivities.

Razeghi et al. [53] focus on the impact of token frequency in pretraining data on LLM performance in arithmetic tasks. A causal framing using the confounding CAT, shown in Figure 3a, captures their hypothesis that token frequency influences accuracy via a backdoor path.

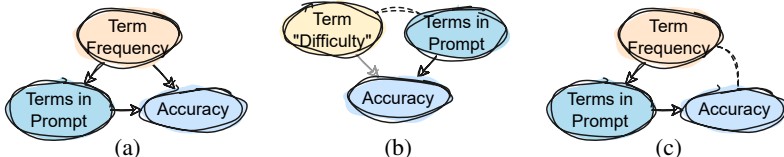

Figure 3: Causal framings of Razeghi et al. [53]: (a) confounding interpretation; (b) imagined alternative setting using a spurious link; (c) a graph combining CATs to avoid unsupported assumptions.

**Alternative Approach**  Here it is instructive to consider a hypothetical project where we design a benchmark to evaluate the math skills of a language model. Much like in Razeghi et al. [53], our questions take the form "What is $n_1$ times $n_2$?" where $n_1$ and $n_2$ are numbers selected by some sampling strategy. However, we do not consider the training dataset of the model at all, and instead we sample numbers uniformly, which effectively removes the causal link between the term frequency and the numbers used in the question. Based on the findings of Razeghi et al. [53], we can expect to find a substantial correlation between the presence of certain numbers in the question and the model's accuracy. To explain the results of our approach, we might phenomenologically posit a new property of numbers called "difficulty" which, the experiments demonstrate significantly affects the model's accuracy, leading to the causal graph in Figure 3b using the spurious link CAT.

**Verifying Causal Assumptions**  Notably, the graph in Figure 3a implies a falsifiable causal relationship between the term frequency and the accuracy which is, strictly speaking, not verified by the experiments of Razeghi et al. [53] which only show a correlation. Therefore, the hybrid causal graph in Figure 3c could be proposed where the correlation is instead due to a hypothesized unknown confounder. This process illustrates how structurally distinct causal interpretations can be proposed to motivate certain experiments or approaches, and then how the results can be used to incrementally refine the causal graph.

### 4.2 Mediation

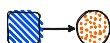

Mediation analysis guides the approaches of mechanistic interpretability [86–90], but it is also useful in the LLMs comprehension of the underlying problem [91], augmentation of language models [92], embedding LLMs within larger programs [93], and the quantification of biases like, gender bias [37].

A common setup for mechanistic interpretability is to study the impact of a specific component, such as an attention head or even a single parameter on the model behavior. Olsson et al. [94] propose that transformers can learn simple, interpretable algorithms called "induction heads," which they hypothesize significantly contribute to in-context learning abilities. While mediation analysis is not explicitly used in their work, we can frame their approach as studying a mediation graph, where the tendency for a given model architecture (stimulus) to exhibit in-context learning (response) is mediated by induction heads. Their six supporting arguments can be interpreted through this causal lens: arguments 1 and 2 establish links between stimulus, mediator, and response through co-occurrence and co-perturbation; argument 3, an ablation study, resembles controlled direct effect estimation; and arguments 4-6 examine the causal influence of the mediator on the response.

This framing also highlights potential limitations, particularly regarding unmeasured confounders that could affect causal interpretations, as the authors' "pattern-preserving" ablation does not fully isolate the induction heads' effect. By considering mediation explicitly, we can better understand the underlying assumptions in their analysis and identify areas for further investigation, such as quantifying the natural indirect effect to understand the full impact of the induction heads on in-context learning abilities. In contrast, Stolfo et al. [87] propose a method for mechanistic interpretability of arithmetic reasoning in LLMs by editing the model's parameters to characterize the information flow in the network.

### 4.3 Spurious Links 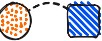

There are several recent projects that use causal models to characterize spurious links in, for example, factual knowledge [95], multi-modal models for fake news detection [96]. To avoid spurious features other projects design strategies to mitigate social biases [59], for finding useful demonstrations in few-shot learning [97] or to control NLP classifiers [98].

Chen et al. [96] develop a causal model to systematically quantify and remove two specific kinds of bias: psycholinguistic (use of emotional language) and image-only (ignoring text features). Note that the assumptions of the causal model address very specific types of bias using both interventional and counterfactual techniques.

Bansal and Sharma [98] presents a particularly interesting case as it addresses the same issue as Gardner et al. [57], but from a causal perspective. They both study the issue of label bias, specifically in "competency problems" [57], where an individual token in the prompt is not indicative of the label, but the model learns to rely on it, usually due to selection bias in the data collection.

The authors of Gardner et al. [57] propose a mitigation strategy based on "local edits" to individual tokens in the prompt to debias the benchmark. Using their statistical framing, the authors prove that the most promising strategy must apply local edits such that the label is flipped precisely half of the time, however, this result not only relies on some unrealistic assumptions, but also provides little insight into why this strategy is effective.

Translating this into a causal framing, we can gain a more general, and more intuitive result. Adopting a similar notation as Gardner et al. [57], the hypothesis can be formalized using the spurious links CAT where a single (text) feature, $X_i$, impacts the model's prediction $Y$ in the presence of the remaining input features $X_{\sim i}$ (stimulus) due to a label bias. Consequently, to remove the effect of $X_i$ on the prediction, we need to design a strategy where the average causal effect (ACE) of $X_i$ on $Y$ conditioned on $X_{\sim i}$ is zero, which is, by definition:

$$\mathbb{E}(Y|X_{\sim i}, do(X_i = x_i')) - \mathbb{E}(Y|X_{\sim i}, do(X_i = x_i)) = 0 \tag{1}$$

where $do(X_i = x_i')$ denotes an intervention (i.e. edit) on feature $X_i$ replacing $x_i$ with some $x_i'$.

Note that Equation 1 is agnostic to the relationship between $X_i$ and $X_{\sim i}$. If we make a "strong independence assumption" as in Gardner et al. [57], then we can directly recover their result that an edit on $X_i$ should flip the label as often as not (i.e. making both terms equal). However, as the authors discuss, this assumption is not realistic as changing a single token may well affect the semantic meaning of the prompt beyond just the label (e.g. replacing "very" with "not" in a movie review). Here the causal framing provides us with a systematic way forward where, we can propose more realistic assumptions, and then use the rules of $do$-calculus to derive a corresponding mitigation strategy, such as the regularization term developed in Bansal and Sharma [98].

In summary, both approaches started with the same objective, but due to the purely statistical treatment, a cumbersome derivation still required an unrealistic assumption severely limiting the applicability of the method. The causal model not only provided a more intuitive motivation for the approach, but also offered a more powerful, principled method opening the door to further analysis, such as quantifying the impact of the label bias (e.g. by estimating the ACE), or controlling the effect, as described in Bansal and Sharma [98].

## 5   Alternative Views

We are hardly the first to point out systematic shortcomings of evaluation methodology, particularly in NLP. One existing perspective focuses on improving the external validity of benchmarks to ensure

that high performance on a benchmark actually translates to improved capabilities in the real world, such as with common sense reasoning [99], or more precisely defining LLMs [100] and how tasks relate to specific cognitive capabilities [66]. Raji et al. [8] argue that the common practice for certain "standard" benchmarks to become proxies for testing complex, high-level abilities, such as natural language understanding (NLU) leads to vague or unreliable results, while Rogers and Rumshisky [46] connect this to a proliferation of low-quality datasets.

Precisely this issue, that "benchmarking for NLU is broken" [9], can be addressed using causality. A causal framing can both provide a versatile way to define the underlying assumptions and design choices of a benchmark, while also offering principled methods for evaluating the benchmark's external validity [101, 102].

In the context of evaluating the reasoning abilities of language models, a natural field to turn to is psychometrics, which has been studying the evaluation of human reasoning abilities for over a century [103]. This direction also coincides with an increasing practice in Natural Language Processing (NLP) to treat language models as agents [104, 105] or subjects in the social sciences [106–108]. Specifically, item response theory [109, 110] holds promise to develop tools to systematically quantify what information about the model's reasoning abilities can be extracted from a benchmark with respect to some population candidate models, and there are some projects applying this framework in the context of NLP [111]. Within the field of NLP there are also notable calls for more holistic evaluation schemes [9, 17, 69] and practical tools for improving the evaluation of language models [10, 112, 113] or even reintroducing principles from linguistic theory [114]. Lastly, there is growing interest in how LLMs acquire and apply causal knowledge [115, 116], including for causal reasoning [117], discovery [118], and even hypothesis generation [119] - this trend aligns with our core message: just as LLMs benefit from explicit causal modeling, so too can the research community.

# 6 Conclusion

The burgeoning research on large models, and, in particular, high-level reasoning tasks, faces a variety of challenges, or *monsters*, to reliably evaluate and improve models. Despite the wide variety of approaches and frameworks that have been developed to tackle these challenges, this variety obscures their shared structural features and recurring issues. By recognizing that monsters can often be effectively formulated in terms of causal assumptions underlying an experimental design or data generation process, we can unify our understanding using the language of causality.

A causal framing aids along several steps of the research process by guiding experimental design, formulating testable hypotheses, and interpreting results. Causal methods enable researchers to gain a clearer lens to understand how variables of interest interact, rather than merely optimizing for predictive performance on an artificial benchmark. We argue that causality offers a path toward deeper scientific insights, more transparent communication of assumptions, and stronger justifications for the conclusions drawn.

One stumbling block to adopting causal methods is that the restrictive assumptions and formalism may seem unapproachable at first. Additionally, researchers may hesitate to commit modeling assumptions to paper where they can be scrutinized. However, data-driven approaches which rely on implicit or vague assumptions along with results that may (inadvertently) be *interpreted* as causal contribute to confusion and unsupported claims, which hinder scientific progress. Causal methods, by contrast, encourage explicit modeling and critical thinking about the mechanisms that underlie empirical observations.

To make causality more accessible and practically applicable, we introduce Common Abstract Topologies (CATs) to faithfully describe the underlying structure of many issues that arise in designing and evaluating ML models. In the case studies, we illustrate how a causal framing can help understand the approach both conceptually in terms of how subtle design choices impact the causal interpretation, as well as technically where the grounded causal machinery can be used to derive a more general, actionable conclusions. Together, these examples demonstrate that even simple causal structures can offer insight into complex evaluation problems, guide experimental design, and refine our understanding of model behavior. We envision CATs as a practical tool, helping researchers quickly identify relevant causal models and choose appropriate inference strategies. Ultimately, causal models encourage more hypothesis-driven research which directly tackle key questions in a principled, transparent way, thereby leading to more robust progress across empirical machine learning.

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
