# OpenReview forum: "Causality can systematically address the monsters under the bench(marks)"
_NeurIPS.cc/2025/Position_Paper_Track — Submitted to NeurIPS 2025 Position Paper Track_

### Official Review · Reviewer_jrjV · 2025-08-08

**Significance:** 3
**Presentation:** 4
**Rating:** 7
**Confidence:** 3

**Summary:**

The authors argue that causality is key when evaluating ML research. In particular benchmark evaluations. The overall argument is that Pearl's causal DAGs provides an unifying language that helps researchers to address the common mistakes and pitfalls of ML evaluation. The authors organize the wide range of issues under common abstract topologies, or CATs. These are mostly well known DAG structures such as confounders and mediators. Nevertheless, for each CAT an extensive review of the literature is provided.

**Strengths:**

Overall, the major strengths are:
- A well organized view of ML evaluation under CATs
- Alternative views are discussed in details
- I particularly liked how the authors acknowledge that current peer review may inhibit the description of causal assumptions in research.

**Weaknesses:**

While I agree with this position paper, my major concern is on the novelty of this position paper. That is, while I view the organization of "monsters" and CATs, as the authors put it, as commendable effort, the importance of causality for evaluation is well known. That is, while this position paper may frame and organize this question for ML benchmarks, econometricians* have argued in favor of causal interpretation of statistical models for decades.

* I understand this is not necessarily Peal causality.

**Questions:**

IMHO, we are at an arms-race like situation with large language models. Whenever a new benchmark that LLMs are unable to solve appear, for instance chain of thought problems, models are quickly updated with training data for that particular task. Most of the cited papers here use causality to find "monsters" in these large models. Following the arms-race, these monsters may be quickly mitigated with an updated training. Do the authors believe that we, humans, will effectively create a systematic approach to continuously measure limitations of large models? If so, how will causality help?

Notice that my question is exactly focused on moving away from the individual monsters/cats and to a more general view. This position paper presents the individual solutions to monsters/cat tied with a unifying argument. However, a systematic guide to help out researchers on their evaluations is not present.

**Alternative Position:**

Yes, and alternative positions are well-considered and addressed by the argument

**Author Identification:**

No.

**Context:**

4

**Details Of Ethics Concerns:**

This paper is a review paper. The literature review presented here by itself does not present any ethical concern. Ethical concerns may exist in cited papers, but this is beyond my evaluation scope.

**Discussion:**

3

**Ethics:**

["NO or VERY MINOR ethics concerns only"]

**Position:**

Yes, the paper argues for or against a position related to machine learning.

**Support:**

4

**Thoroughness:**

2

---

### Official Review · Reviewer_Spky · 2025-08-13

**Significance:** 4
**Presentation:** 3
**Rating:** 7
**Confidence:** 4

**Summary:**

The authors argue that an effective and reliable evaluation is essential for advancing empirical machine learning. Nevertheless, benchmarks suffer from multiple biases, artifacts, or leakages, undermining their reliability. The authors propose a causal framework to address such challenges and identify several Common Abstract Topologies that could be used to gain insights into the reasoning abilities of large language models.

**Strengths:**

(S1)- The authors introduce an original perspective on the topic, diagnosing typical issues found in benchmarks and proposing a catalogue of Common Abstract Topologies rooted in causality to mitigate them.

(S2)- The authors motivate their approach in multiple use cases.

(S3)- Through the proposed framework, the authors understand that many assumptions made at different benchmarks would be made explicit, leading to better communication, transparency, and overall benchmark quality.

**Weaknesses:**

(W1)- The title is deceiving: it seems to address benchmarks in general, while the paper focuses only on benchmarks related to LLMs.

(W2)- While the idea is valuable, it is not clear how the whole benchmark specification should be done, and the benchmark implemented and validated against that specification. In fact, such an approach would require a library of CATs, means to implement them as a benchmark, and some validation tool to validate whether the benchmark complies with the specification.

**Questions:**

(Q1)- Did the authors consider some means to scan existing benchmarks and automatically propose CATs along with possible gaps that must be validated with human intervention to ensure adequate coverage vs. the original benchmark formulation?

(Q2)- While in the case studies, the authors reference several sources from the literature, did they execute some experiments to understand (a) to what extent the proposed strategies are helpful to the goal under consideration, and (b) what would be the impact of realizing the proposed approach in reality? To understand the potential impact providing some limited results and validation.

(Q3)- Could a similar approach generalize to other kinds of benchmarks, and the idea described in this paper provide the ground for a more general approach on how to ensure a certain level of benchmarking quality?

(Q4)- The authors are concerned with the reasoning process of a model to understand whether the reasoning is valid to arrive at a certain result. Did they consider a unified approach of explainability and causal assessment as part of the benchmark specification?

**Alternative Position:**

Yes, and alternative positions are well-considered and addressed by the argument

**Author Identification:**

No.

**Context:**

3

**Discussion:**

4

**Ethics:**

["NO or VERY MINOR ethics concerns only"]

**Position:**

Yes, the paper argues for or against a position related to machine learning.

**Support:**

3

**Thoroughness:**

5

---

### Official Review · Reviewer_219h · 2025-08-13

**Significance:** 3
**Presentation:** 3
**Rating:** 6
**Confidence:** 3

**Summary:**

This position paper argues that the language and tools of causality offer an ideal framework to systematically address the myriad of challenges plaguing machine learning evaluation. The authors contend that many issues, such as biases, artifacts, and spurious correlations, are often treated with ad-hoc solutions but share underlying causal structures. To make causal modeling more accessible, they introduce "Common Abstract Topologies" (CATs) as simple, intuitive templates for confounding, mediation, and spurious links. Through a series of case studies focused on LLM reasoning, the paper demonstrates how this causal framing can clarify assumptions, motivate principled experiments, and lead to more generalizable insights than purely statistical approaches. Ultimately, the paper advocates for a shift towards more explicit, hypothesis-driven research grounded in causal principles to foster more robust and reliable scientific progress.

**Strengths:**

1. The paper unify a diverse problems described with vague terminology ("shortcuts," "cheating," "artifacts") under the framework of causality. This reframing into identifiable causal structures is a sound conceptual contribution.
2. The introduction of Common Abstract Topologies (CATs) is a brilliant device for making the often intimidating formalism of causal inference more accessible. By providing simple, recognizable graphical patterns (Table 1), the authors offer a practical starting point for researchers to begin incorporating causal thinking into their work.
3. The case studies (Section 4) provide concrete evidence for the paper's claims. The analysis of how a causal perspective yields a more general and intuitive solution for the label bias problem is particularly persuasive, directly demonstrating the technical superiority of the causal approach over a statistical one.

**Weaknesses:**

1. The paper aims to make causality more accessible, but correctly applying causal inference remains a highly skilled task. The leap from identifying a CAT to deriving the correct estimand using do-calculus (as in Equation 1) is non-trivial, and the paper may be slightly optimistic about how easily the broader community can adopt these methods without specialized training.
2. The CATs and case studies deal with well-defined, relatively low-dimensional problems (e.g., the effect of a single token or a specific model component). It is less clear how this framework would scale to diagnosing issues in the entire pre-training data-generating process of an LLM, which involves trillions of tokens and countless unobserved confounding factors.

**Questions:**

Your case studies are compelling but focus on well-circumscribed research questions. How would you propose applying the CATs framework to diagnose potential "monsters" in a far more complex and opaque setting, such as the web-scale data curation pipeline used for pre-training a foundation model?

**Alternative Position:**

Yes, and alternative positions are well-considered and addressed by the argument

**Author Identification:**

No.

**Context:**

3

**Discussion:**

3

**Ethics:**

["NO or VERY MINOR ethics concerns only"]

**Position:**

Yes, the paper argues for or against a position related to machine learning.

**Support:**

4

**Thoroughness:**

3

---

### Note · Authors · 2025-08-29

**1-10 Additional Comments:**

Thank you for the 3 quality reviews. From their constructive and thought-provoking feedback, it is evident that the reviewers carefully considered our submission.

**1-11 Submit Again:**

Probably yes

**1-1 Submission Process:**

3

**1-2 Next Year:**

More reviews and a discussion period would be nice.

**1-3 Future Development:**

There should be a discussion period (if not a full rebuttal period) to give authors a chance to better understand and even incorporate the feedback of reviewers into their submission.

**1-4 Interest:**

["Panel discussions with other position paper authors", "Structured debates on controversial topics"]

**1-5 Thoughtful:**

8

**1-6 Supportive:**

9

**1-7 Technical Aspects Versus Position:**

7

**1-8 Gate Keeping:**

8

**1-9 Camera Ready Changes:**

While we are pleased at the positive reception by the reviewers, several reviewers raised some interesting questions that we would like to include in some form in the camera ready. In particular, reviewer jrjV raised an interesting question regarding the underlying motivation of our proposed framework and whether it can help in the face of an apparent "arms race" where new issues in LLMs are patched ad hoc without systematic understanding. A question by reviewer Spky about a unified approach of explainability in LLM reasoning also raised an interesting point in how we can connect our work to the burgeoning field of causal reasoning in LLMs.

**3-1 Review Response1:**

219h

**3-2 Reaction To Review1:**

We thank the reviewer for their thoughtful and supportive feedback.

Although, we partly share the reviewers' concerns about accessibility and scalability, we would like to emphasize two points: (1) while it is certainly too optimistic to expect explicitly derived estimands to become standard practice in ML papers, we believe that many of the conceptual insights from causal inference can serve as a stepping stone to encourage more rigorous approaches where appropriate. (2) On the issue of scalability, we acknowledge the challenges in attempting to construct (even conceptually) a causal model for foundation model pre-training pipelines, as you suggest. While a comprehensive causal model may be infeasible, most projects aim to identify/mitigate some more specific aspect (or monster) of a complex system, where our CAT framework can offer guidance with a suitable level of abstraction. For instance, rather than attempting to model the entire data pipeline, researchers could use the CAT framework to analyze specific components, such as the impact of data augmentation strategies or the role of particular evaluation metrics. This targeted approach allows for more manageable causal inquiries within the broader context of LLM training.

Overall, the review was pleasantly constructive. It engaged directly with our position offering support as well as balanced critique, which we will include in the camera-ready version.

**3-3 Review Response2:**

Spky

**3-4 Reaction To Review2:**

We thank the reviewer for their insightful feedback. We are encouraged that they found our perspective original and the potential for improved transparency valuable. We address some of the weaknesses and questions below.

**(W1, W2, Q1) Automated CAT Discovery:** We will clarify in the introduction that our case studies focus on LLMs. We agree that developing practical validation tools based on our framework is an exciting direction for future work. Automated CAT discovery is an interesting suggestion for future research. However, we anticipate a key challenge would be reliably extracting the contextual knowledge required for a sound automated causal analysis.

**Q4: Unifying with Explainability:** This question touches on an intriguing connection between our work here and the growing subfield of *causal* reasoning in LLMs. Indeed we believe there is a deep connection between improving the reasoning of LLMs (such as through explainability) and understanding the causal structure of the underlying data generative process. Here we focus on the latter for improved (human) research communication and benchmark design, however making such causal assumptions accessible to an LLM will likely enhance its reasoning abilities.

Thank you again for the detailed and thought-provoking feedback. We will incorporate these points to improve the camera-ready version of our paper.

**3-5 Review Response3:**

jrjV

**3-6 Reaction To Review3:**

We thank the reviewer for their supportive and insightful feedback. We're glad you found our organization of ML evaluation issues well-argued and the presentation excellent.

**On Novelty:** We agree that the core principles of causality are well-established in fields like econometrics. Our primary contribution is not to reinvent these ideas, but to translate, organize, and systematize them for the machine learning community. While the concepts may be known to causal inference experts, they are not part of the standard toolkit for most of the ML community. Our paper aims to bridge this gap by providing a common language to make these powerful ideas more accessible and actionable for ML benchmark design and evaluation.

**On the LLM "Arms Race":** Your question about the "arms race" and the need for a systematic approach gets to the heart of our paper's motivation. We agree that the current cycle of finding and patching individual "monsters" is reactive and unsustainable. Our approach is to develop tools that encourage improved understanding and communication of proposed methods, rather than seeking a framework that directly resolves the "monsters." In this everchanging landscape of new training protocols and failure modes, causality alone will not solve these issues (as we discuss in the introduction), however, we argue, a more explicitly causal lens can help by systematizing the design choices and their consequences. Thus, we share your concerns about the need for a more general view.

Thank you again for engaging so thoughtfully with our position. Your feedback will help us sharpen our argument for the camera-ready version.

---

### Meta-Review · Area_Chair_F1xU · 2025-09-06

**Rating:** 7
**Confidence:** 3

**Strengths:**

– Unifies diverse problems under the framework of causality.
– Introduces CAT as an accessible device
– Provides concrete evidence for the claims made.
– Original perspective
– The proposed framework helps make assumptions explicit.
– Alternative views discussed in detail.

**Weaknesses:**

– Though the proposal aims to be accessible, adopting the proposed methods still requires specialization.
– It is unclear how the framework would scale to real-world models.
– Title is not representative.
– Unclear how the framework can be applied in practice.
– Novelty

**Questions:**

219h’s question regarding practical application
Spky’s questions regarding automated proposal of CATs, impact, more general benchmarks, and unified approach fo explainability and causal assessment..
jrjV’s question regarding the arms race scenario.

**Ethics:**

No.

**Thoroughness:**

3

---

### Decision · Program_Chairs · 2025-09-26

Reject